# Potential Production of Theranostic Boron Nitride Nanotubes (^64^Cu-BNNTs) Radiolabeled by Neutron Capture

**DOI:** 10.3390/nano11112907

**Published:** 2021-10-30

**Authors:** Wellington Marcos Silva, Helio Ribeiro, Jose Jaime Taha-Tijerina

**Affiliations:** 1Departamento de Química, Universidade Federal de Minas Gerais, Avenida Presidente Antônio Carlos, 6.627, Belo Horizonte 31270-901, MG, Brazil; wellingtonmarcos@yahoo.com.br; 2Escola de Engenharia, Mackenzie Presbyterian University, Rua da Consolação 896, São Paulo 01302-907, SP, Brazil; helio.ribeiro1@mackenzie.br; 3Departamento de Ingeniería, Universidad de Monterrey, Av. Ignacio Morones Prieto 4500 Pte., San Pedro Garza García 66238, NL, Mexico; 4Engineering Technology Department, University of Texas Rio Grande Valley, Brownsville, TX 78520, USA

**Keywords:** theranostic nanomaterials, boron nitride, neutron capture reaction, nuclear medicine

## Abstract

In this work, the radioisotope ^64^Cu was obtained from copper (II) chloride dihydrate in a nuclear research reactor by neutron capture, (^63^Cu(n,γ)^64^Cu), and incorporated into boron nitride nanotubes (BNNTs) using a solvothermal process. The produced ^64^Cu-BNNTs were analyzed by TEM, MEV, FTIR, XDR, XPS and gamma spectrometry, with which it was possible to observe the formation of^64^Cu nanoparticles, with sizes of up to 16 nm, distributed through nanotubes. The synthesized of ^64^Cu nanostructures showed a pure photoemission peak of 511 keV, which is characteristic of gamma radiation. This type of emission is desirable for Photon Emission Tomography (PET scan) image acquisition, as well as its use in several cancer treatments. Thus, ^64^Cu-BNNTs present an excellent alternative as theranostic nanomaterials that can be used in diagnosis and therapy by different techniques used in nuclear medicine.

## 1. Introduction

The discovery and development of new materials are often the catalysts for technological advances, particularly when they can be applied to various areas of research. A historic milestone in the search for new materials occurred in 1995 when boron nitride nanotubes (BNNTs) emerged as a key material in nanotechnology science [1]. BNNTs have cylindrical structures (one-dimensional—1D), formed only by atoms of boron (B) and nitrogen (N), with diameters in the order of nanometers and lengths in the order of microns [2]. Similar to single-wall carbon nanotubes (SWCNT), depending on the angle *θ* in which the sheet is rolled, nanotube structures are formed with armchair (*θ* = 30°), zig-zag (*θ* = 0°) and chiral (0 < /*θ*/ < 30°) forms [3]. Otherwise, the multi-walled BNNTs are formed by several sheets of hexagonal boron nitride (h-BN) concentrically wrapped. Both BNNTs have excellent chemical, physical and mechanical properties [4,5,6,7,8] and a band gap of approximately 5.5 eV [3]. Several methods have been used for the synthesis of BNNTs [5]; however, chemical vapor deposition (CVD) is the most used method and requires a simple apparatus which produces BNNTs with excellent structural quality [9].

The interest in BNNT-based technologies in recent years can be measured by the large number of scientific works that have been published, in addition to the increase in large-scale production [10]. Within the class of nanostructured materials, the BNNTs have great potential for several biomedical applications. Some early studies demonstrated that nanotubes did not show toxicity at concentrations below 50 µg/mL [11]. Furthermore, they favor the reuptake of molecules into the cell interior and can be functionalized with different biological protein epitopes [12]. Recently, BNNTs have been used as nanovectors for DNA, drugs and radioisotopes, and as boosters for biomaterials. In 2012, Soares et al. [13] used BNNTs radiolabeled with ^99m^Tc to investigate the cell-distribution behavior *in vivo* through a process of passive accumulation in solid tumors. Diverse studies applying BNNTs to cancer treatment have been reported. For example, when linked to target molecules, BNNTs could be used as therapeutic agents capable of killing cancer cells by boron neutron capture therapy. This medical approach is generally applied in brain cancer treatments, and it is based on the capture of the neutron reaction ^10^B (n,α) ^7^Li, where a ^10^B atom captures a low-energy thermal neutron and then decays to produce ^4^He (alpha particles) and ^7^Li, resulting in a dense ionizing radiation which is capable of destroying the cells where the reaction takes place [12]. Another potential application of BNNTs is in diagnostic medicine. In this sense, BNNTs doped with rare earth beta-emitters with short half-lives, such as ^153^Sm and ^159^Gd, can also be used as radioisotopes for imaging [14].

In this context, nanotechnology has revolutionized so-called traditional medicine by introducing novel concepts and methods that had never been imagined. Thus, nanomedicine has improved the diagnosis of various diseases through techniques based on magnetism or nuclear reactions with different electronic devices, using biosensors or radioisotope-doped nanomaterials. In this way, the study of a more accurate diagnostic method using novel technologies is as relevant a goal as the prevention and treatment of oncological diseases. Therefore, a class of new nanomaterials, in which boron nitride nanotubes (BNNTs) stand out, has been the target of studies that have led to an understanding of the correlation between their structure and properties, which enables their use in diagnostic medicine. Due to their empty internal spaces, BNNTs can be filled by different chemical species, such as enzymes, noble metals, rare earths, and radioisotopes, especially copper-64 (64-Cu), which allows this type of material to be applied as a biological marker and in diagnostic medicine. For instance, copper-64 (T_1/2_ = 12.7 h; β+, 0.653 MeV (17.8%); β^-^, 0.579 MeV (38.4%)) has decay characteristics that allow it to be applied to obtain images of positron emission tomography (PET-scan) and in cancer-directed radiotherapy. Copper, for instance, has already well-established coordination chemistry that allows its reaction with an extensive variety of chelating systems that could potentially be linked to peptides and other interesting biological molecules such as antibodies, proteins, and nanoparticles. Its specific half-life expands the ability to image molecules of various dimensions, mainly including the slower compensating proteins and nanoparticles. Due to the versatility of applications of 64-Cu, a significant increase in scientific and technical publications has been seen over the last 2 decades, mainly in PET-scan imaging, but also in targeted cancer radiotherapy.

Thus, this work aimed to synthesize and characterize ^64^Cu-BNNTs with appreciable properties that suggest numerous multifunctional applications, with advantages for cancer diagnosis and therapy, such as: (i) increased bioavailability; (ii) reduction in systemic adverse effects, thereby increasing patient comfort and adherence to treatment; (iii) improved osteogenic differentiation response promoted by the ^64^Cu-BNNTs system and targeting of tumor cells, among others. It is also important to mention that the combination of ^64^Cu-BNNTs has not yet been reported in the literature.

## 2. Experiment

### 2.1. Raw Materials

Copper (II) chloride dihydrate (99.999%), iron (III) oxide nano powder (<50 nm particle size) and amorphous boron powder (≥95%) were obtained from Sigma Aldrich Brazil-Ltda, Sao Paulo, Brazil (CAS Number 10125-13-0) and used as received.

### 2.2. Synthesis and Purification of Boron Nitride Nanotubes

BNNTs were processed from mixing amorphous boron and iron (III) oxide powder (ratio 0.02) in a horizontal tubular reactor. This reactor consisted of an alumina with an inlet and outlet for the flow of ammonia and nitrogen gases. The synthesis was carried out under a NH_3_/N_2_ atmosphere at a 150/20 sccm (standard cubic centimeters per minute) flow rate with a heating rate of 10 °C min^−1^ from room temperature up to 1200 °C. An isotherm was maintained for 2 h. After this step was completed, the reactor was cooled down to room temperature under a N_2_ atmosphere.

The synthesized BNNTs were purified using sulfuric and nitric acids in the ratio of 3:1, respectively. The reaction mixture was kept under stirring and reflux conditions at 80 °C for 2 h, followed by the filtration process. The resulting solid was washed with deionized water and oven-dried for 4 h at 110 °C. In this process, hydroxyl groups (-OH) were introduced into the structure of the tubes.

#### 2.2.1. Activation Process of ^64^Cu Radioisotope

The radioisotope ^64^Cu was obtained by neutron activation of the copper (II) chloride dihydrate sample in a nuclear research reactor (TRIGA Mark-1) at CDTN (Belo Horizonte, Brazil) by the neutron capture reaction ^63^Cu(n,γ)^64^Cu. The irradiation was performed on 20 mg samples over 8 h under a thermal neutron flux of 6.6 × 1011 cm^−2^s^−1^. The theoretical induced activities were estimated according to the research of Zangirolami et al. [15]. The calculations were carried out while considering the amount of Cu in the sample and using the thermal neutron capture cross-sections as a reference, in accordance with an IAEA (International Atomic Energy Agency) publication [16].

#### 2.2.2. Incorporation of Cu and ^64^Cu to the BNNT Samples

The BNNT (100 mg) sample was dispersed in anhydrous ethanol. With the aid of an autoclave with a polytetrafluoroethylene (PTFE) vessel, the Cu and ^64^Cu radioisotope were incorporated into the BNNTs. The incorporation reaction was carried out in an oven at a temperature of 180 °C for two hours. After this period, the material was cooled to room temperature and filtered. The radiochemical purity of the sample was assessed by gamma spectroscopy, using an HP-Ge detector (Ortec Ametek, Oak Ridge, TN, USA) with 25% efficiency, and analyzed using the Canberra Genie 2000 software, Meriden, CT, USA [17]. The evaluation of the specific activity was carried out using a CRC 15R activimeter that had been previously calibrated for copper-64 emission. Figure 1a,d schematically show all stages of ^64^Cu-BNNT production.

## 3. Characterization

FTIR measurements of the BNNT and Cu-BNNT samples were performed with a Bruker model Vertex 70v instrument (Belo Horizonte, Brazil). The spectra were collected in ATR mode with 64 accumulations, a resolution of 4 cm^−1^, and in the 4500–300 cm^−1^ region in transmission mode and then were systematically adjusted; baseline corrections were considered for this analysis. An ultima IV Rigaku Diffractometer with Cu-Kα radiation was employed to study the main crystalline phases in the synthesized BNNT and Cu-BNNT samples (Belo Horizonte, Brazil). The Bragg’s angle values were measured in the 10–80° range, with a scanning rate of 0.02° min^−1^. XPS spectra were obtained using monochromatic Al Kα radiation (1486.6 eV) with an electron energy analyzer (Specs, Phoibos-150) that enabled high-energy resolution and an excellent signal-to-noise ratio (Belo Horizonte, Brazil). The signal of adventitious carbon (C 1 s at 284.6 eV) was used to correct the binding-energy scale of the survey and the high-resolution spectra. High-resolution spectra in the regions of interest were fitted assuming its shape as a convolution of Lorentzian and Gaussian functions of different components, and the background contribution was removed by the Shirley method [18,19]. SEM analysis was performed with a Carl Zeiss Field Emission Scanning Electron Microscope, model sigma VP (Belo Horizonte, Brazil), operating in vacuum with an electron-beam-acceleration voltage between 5 and 30 kV. The BNNT and Cu-BNNT powders were deposited directly onto the carbon tape. The Transmission Electron Microscopy (TEM) images were obtained on a FEI TEM-LaB6 TECNAI G2 microscope (Belo Horizonte, Brazil), with a tungsten-filament electron gun operating at 200 kV. Samples were dispersed in acetone for 30 min using a water bath sonicator and one drop was deposited onto a 200-mesh holey carbon–copper grid. The activity of the ^64^Cu-BNNTs after irradiation was obtained from the gamma spectrum, using an HP-Ge detector (Belo Horizonte, Brazil), with a nominal efficiency of 25%, and the Canberra Genie 2000 software.

## 4. Results and Discussion

The FTIR spectrum was obtained in order to identify the vibrational modes in the BNNT samples (Figure 2). The absorption peaks between 3400 and 3200 cm^−1^could be attributed to the vibrational modes of the hydroxyl groups (-OH) from water molecules adsorbed on the sample surface [20,21]. However, it could also be attributed to the presence of copper hydroxide. The region between 2000 and 60 cm^−1^ has several peaks (Figure 2b,c). The well-known longitudinal (LO) vibrations along the axis resonate sharply around 1369 cm^−1^, and a second signal (1545 cm^−1^) appears for tangential (T) circumferential in-plane modes (νB-N). These T modes should be dependent on the diameter (curvature) but seem to only be visible in highly pure, crystalline BNNTs [22]. Another typical absorption peak for BNNTs is located around 790 cm^−1^ and is related to out-of-plane B-N-B bending (δB-N-B) vibrations [20,21,23]. In both spectra, the peaks between the 1100 and 880 cm^−1^ regions give an account of the anti-symmetrical and symmetrical stretching vibrations of B-O bonds in BO_3_ and BO_4_ groups formed from B-OH, and peaks at 701.3, 685.8 and 453.3 cm^−1^ are assigned to the bend vibrations of B-O bonds in BO_3_ and BO_4_ groups [24]. The peak at 426.0 cm^−1^ is assigned to the stretching vibrations νCu(II)-O of copper oxide CuO [25].

The XRD of BNNTs and Cu-BNNTs is shown in Figure 3. An intense peak close to 2*θ* = 26.65° (Figure 3a) corresponds to the plane (002) and is attributed to the main peak of the h-BN structure. Peaks assigned to h-BN are also observed at 2*θ* = 41.78°, 42.81°, 50.16°, 55.09°, 59.40° and 76.05^◦^, which correspond to the (100), (101), (102), (004), (103) and (110) planes, respectively [14,23,26]. After the introduction of Cu nanoparticles to the BNNTs, new diffraction peaks were observed (Figure 3b),so the region between 30° and 80° was highlighted. The presence of CuO and Cu_2_O nanoparticles were identified at 36.89°, 39.71° and 65.3°, which may have occurred due to the exposure of the nanoparticles to the surrounding environment during characterization [27]. The characteristic diffraction peaks of copper nanoparticles located at 32.42° and 44.81° were observed. They correspond to the (110) and (200) crystallographic planes of face-center cubic (fcc), respectively [27,28]. Debye–Scherrer’s equation, i.e., D = 0.9 × λ/(β × cos*θ*), was used to calculate the size of copper nanoparticles, where D represents crystalline size, 0.9 is Scherrer’s constant, λ is the wavelength of the X-ray, β is the full width at the half-maximum of the diffraction peak (FWHM) and *θ* represents Bragg’s angle [27,29]. The calculations were performed using the mean values of the FWHM of the peaks, with 2*θ* of 32.4° and 44.8°. The average size of the Cu nanoparticles was 16 nm. This is a dimension in which nanoparticles can penetrate tumor cells through the Increased Permeability and Retention Effect (EPR), thus they can be used as a theranostic nanomaterials. 

Figure 4 shows the XPS survey of the samples. In all the survey spectra, only the presence of B (190.10 eV), N (398.12 eV), C (284.49 eV) and O (532.51 eV) was identified. The C and O signals were also identified on the BNNT surface. The presence of C is related to the surface contamination that usually occurs during the preparation process and to the exposure of the specimens to air, and the presence of O is due to the purification process and is commonly observed in XPS measurements. The presence of F (689.51 eV) and Si (102.53 eV) is due to residues from the vessel that was used for synthesis and the Cl (198.53 eV) comes from the reagent CuCl_2_·2H_2_O that was used for the synthesis of copper nanoparticles.

Table 1 shows the data obtained from the XPS survey. The stoichiometry rate of boron and nitrogen atoms (B:N) is confirmed by the peak areas of the XPS survey spectra. The obtained value for the B:N ratio is 1.09. A similar result was obtained by Silva et al. and Juan Li et al. [21,30]. According to these studies, the non-stoichiometric BN nanotubes present an excess of B atoms, and oxygen-doping in the h-BN network leads to the formation of ternary BN_x_O_y_ species [31]. Hydroxyl groups increase in modulus to negative charges on the tubes’ surfaces. As copper nanoparticles have positively charged surfaces, we believe that an electrostatic interaction occurs between the two nanostructures.

The B 1s, N 1s and O 1s core-level photoemission spectra for all samples are shown in Figure 5. The B 1s peak (Figure 5a,b) at 190.1 eV and N 1s peak (Figure 5c,d) at 398.0 eV correspond to the B–N bonding, matching the BE values reported for bulk h-BN [21]. The component at 188.3 eV (Figure 5a,b) is attributed to B bonded to Fe from the catalyst (Fe_2_B) [32]. A minor contribution of boron oxide (B_2_O_3_) was also identified at 192.4 eV (Figure 5a). After the synthesis of Cu nanoparticles, two new contributions were observed at 200.4 and 197.0 eV (Figure 5b); the first is attributed to Cl II and the second to the presence of metallic copper (Cu 0). In addition, the binding energies at 396.6 eV and 400.0 eV are attributed to B–N–B bonding [33] and O-B-N bonding [14], respectively. The O 1s spectra are also shown in Figure 5e,f. Both samples show peaks at 535.5, 532.9 and 530.2 eV, respectively. The first is associated with oxygen atoms bonded to O-H from the purification process, the second is characteristic of B–O bonds (B_2_O_3_) and the third is associated with oxygen atoms bonded to Fe-O (Fe_2_O_3_), which is related to the synthesis process [26].

Figure 6 depicts the high-resolution spectrum XPS for Cu-BNNTs. The dominant Cu 2p_3/2_ peaks at 933.8 and Cu 2p_1/2_ at 953.6 eV, which demonstrate the successful reduction of CuCl_2_, can be assigned to metallic Cu(0), whereas the small peak at 936.2 eV can be assigned to Cu(II). Meanwhile, the peaks of Cu 2p_1/2_ at 955.9 eV, in combination with the satellite peak at 944.3 eV, are typical characteristics of CuO, implying the uniform surface oxidation of Cu nanoclusters that were exposed to air under ambient conditions [34,35]. Cu(0) and Cu(I) are hard to differentiate since they have a ~0.3 eV separation in binding energy, whereas Cu(0) and Cu(II) have more than a 2eV separation. The Cu 2p_5/2_ peak around 958.1 eV indicates a Cu(II) oxidation state [36,37].

The morphological characteristics of the produced samples were studied using SEM and TEM microscopy, shown in Figure 7 and Figure 8, respectively. It was observed in both samples that the produced BNNTs have non-uniform lengths and diameters. This feature is due to the nanotubes entanglement during their growth process, which is very common during the synthesis process. This nanotube characteristic was also observed in our previous work [23].

TEM images are shown in Figure 8a–f. The highlighted regions in Figure 8a,c,e are shown in high-resolution images in Figure 8b,d,f. All images represent typical nanotubes with defined internal channels and external walls that are structurally well-organized [14].

When comparing Figure 8a,b with Figure 8c–f, particles with higher electron absorptions, appearing as darker sites isolated from each other, can be associated with the presence of an electron-conducting nanostructure, indicating the presence of metallic Cu [33]. It is expected that combining these metal nanostructures with BNNTs can improve the stability of the nanoparticles within the solution, allowing their application as theranostic nanomaterials. After increasing image resolution (Figure 8d–f) it was observed that Cu nanoparticles are present in great numbers on the tube surfaces and inside the channels. Referring to the scale bar, it is important to note that the size of nanoparticles is close to 16 nm, as determined by the Debye–Scherrer´s equation.

The ^64^Cu nanoparticles, synthesized from the high-purity CuCl_2_·H_2_0 by using a solvothermal method, produced stable and contaminant-free radioisotopes, as shown in the gamma spectrum in Figure 9.

The pure photopeak of the ^64^Cu-BNNTs is 511 keV [38,39]; this energy is compatible with gamma rays for the obtention of images by Photon Emission Tomography (PET-scan), and for cancer treatment due to its β-emissions, with an energy of 579 keV. This result illustrates that the ^64^Cu-BNNTs can be used as a potential nanomaterial that is able to produce images as well as promote several cancer treatments.

## 5. Conclusions

The ^64^Cu nanostructures were incorporated within the BNNTs structure by a solvothermal method which produces stable and contaminant-free radioisotopes. Using XDR data and Debye–Scherrer’s equation, it was possible to determine that metallic Cu nanoparticles have sizes of about 16 nm. The TEM images showed that the BNNTs are structurally well-organized, presenting Cu nanoparticles in their internal channels with an even distribution on their surfaces. The ^64^Cu nanoparticles in the BNNTs also showed a pure photoemission peak of 511 keV, which is characteristic of gamma radiation. These results corroborate the fact that the studied system has high potential to be used in nuclear medicine as a theranostic material. However, this subject needs to be further explored.

## Figures and Tables

**Figure 1 nanomaterials-11-02907-f001:**
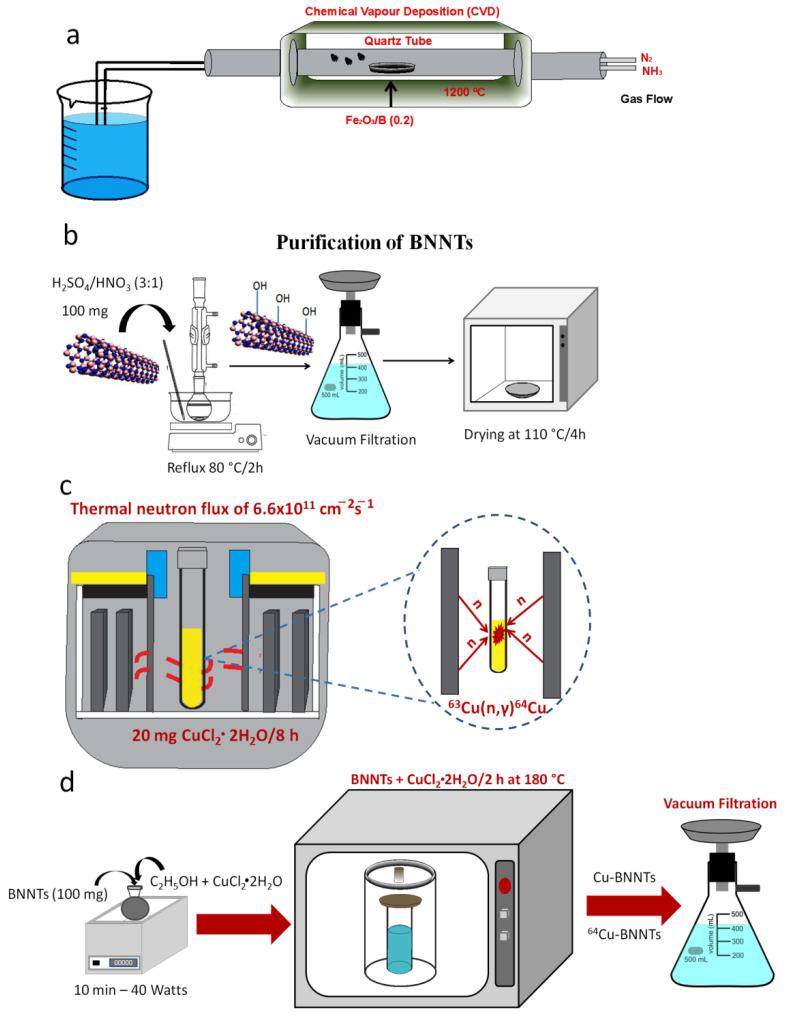
Schematic representation of synthesis of the BNNTS (**a**), purification process (**b**), activation process of CuCl_2_·2H_2_O to obtainment of ^64^Cu radioisotope (**c**) and incorporation of Cu and ^64^Cu into BNNTs samples (**d**).

**Figure 2 nanomaterials-11-02907-f002:**
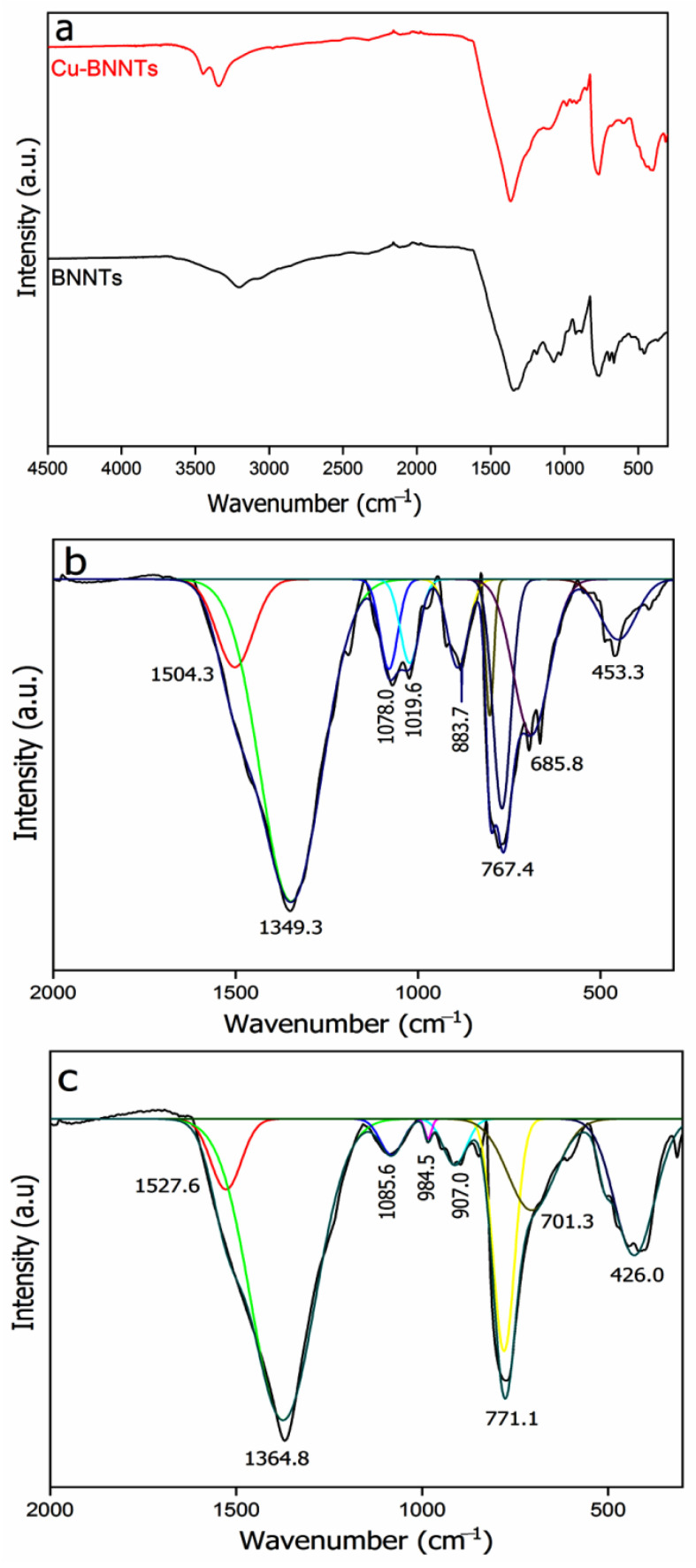
Infrared spectra of (**a**) BNNTs and Cu-BNNTs in the region between 4500 and 60 cm^−1^. (**a**) Highlighted regions between 2000 and 60 cm^−1^ for (**b**) BNNTs and (**c**) Cu-BNNTs.

**Figure 3 nanomaterials-11-02907-f003:**
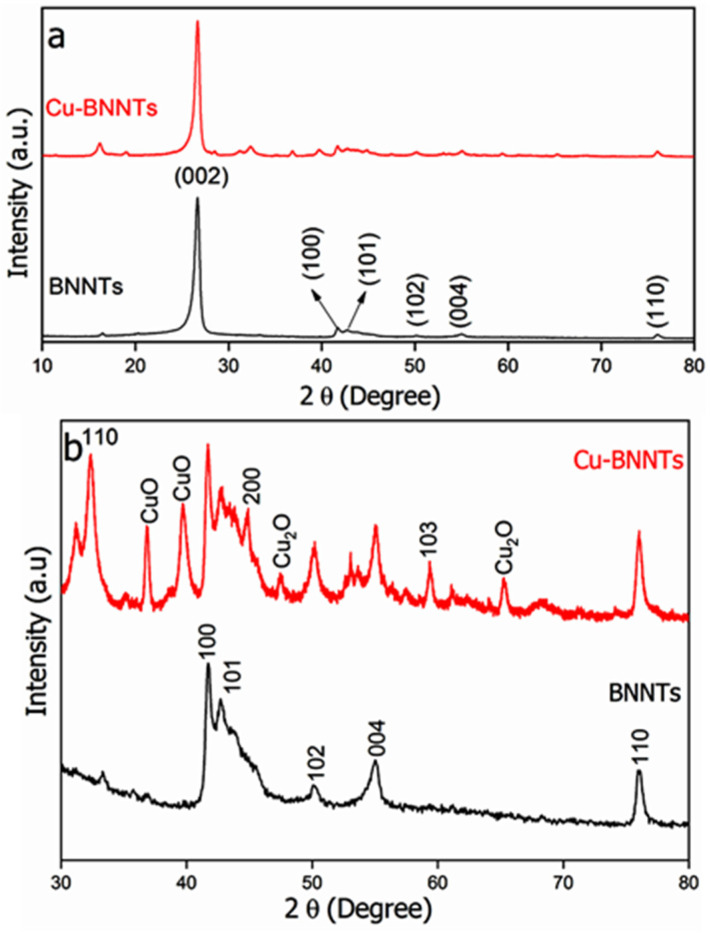
XRD of BNNTs and Cu-BNNTs in the region between 10° and 80° (**a**). Highlighted regions between 30° and 80° (**b**).

**Figure 4 nanomaterials-11-02907-f004:**
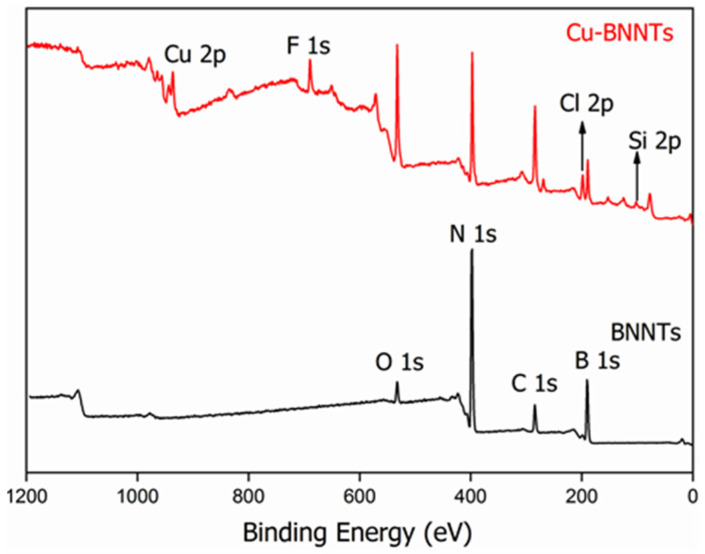
Survey XPS spectra for BNNTs and Cu-BNNTs.

**Figure 5 nanomaterials-11-02907-f005:**
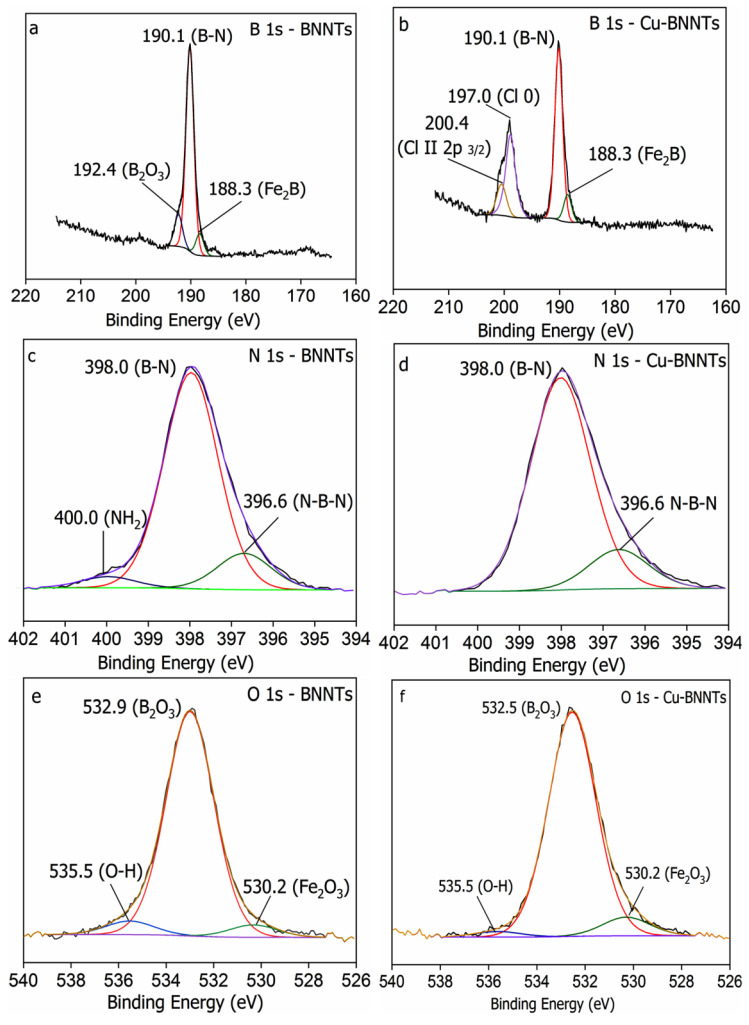
B 1s, N 1s and O 1s high-resolution XPS spectra for (**a**,**c**,**e**) BNNTs, (**b**,**d**,**f**) Cu-BNNTs.

**Figure 6 nanomaterials-11-02907-f006:**
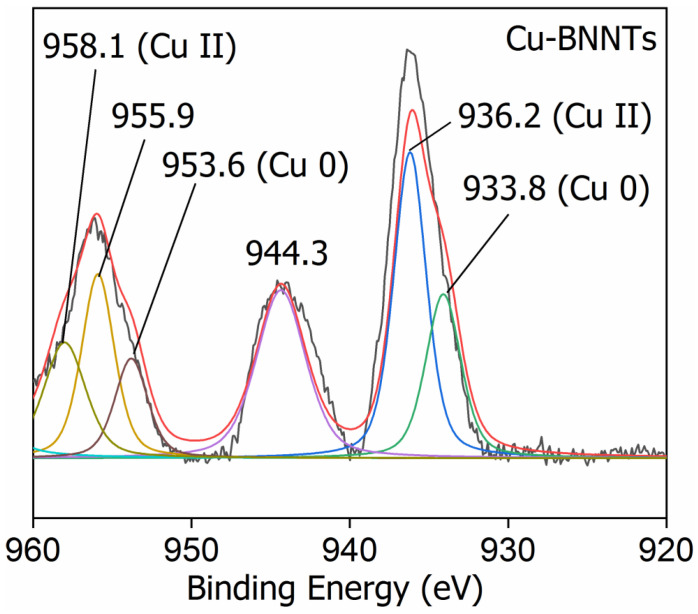
High-resolution spectrum XPS for Cu-BNNTs.

**Figure 7 nanomaterials-11-02907-f007:**
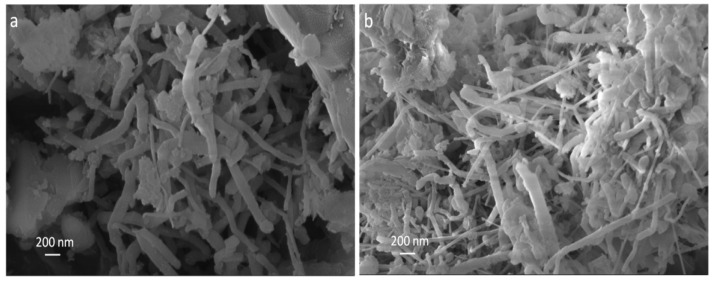
SEM images of BNNTs (**a**) and Cu-BNNTs (**b**).

**Figure 8 nanomaterials-11-02907-f008:**
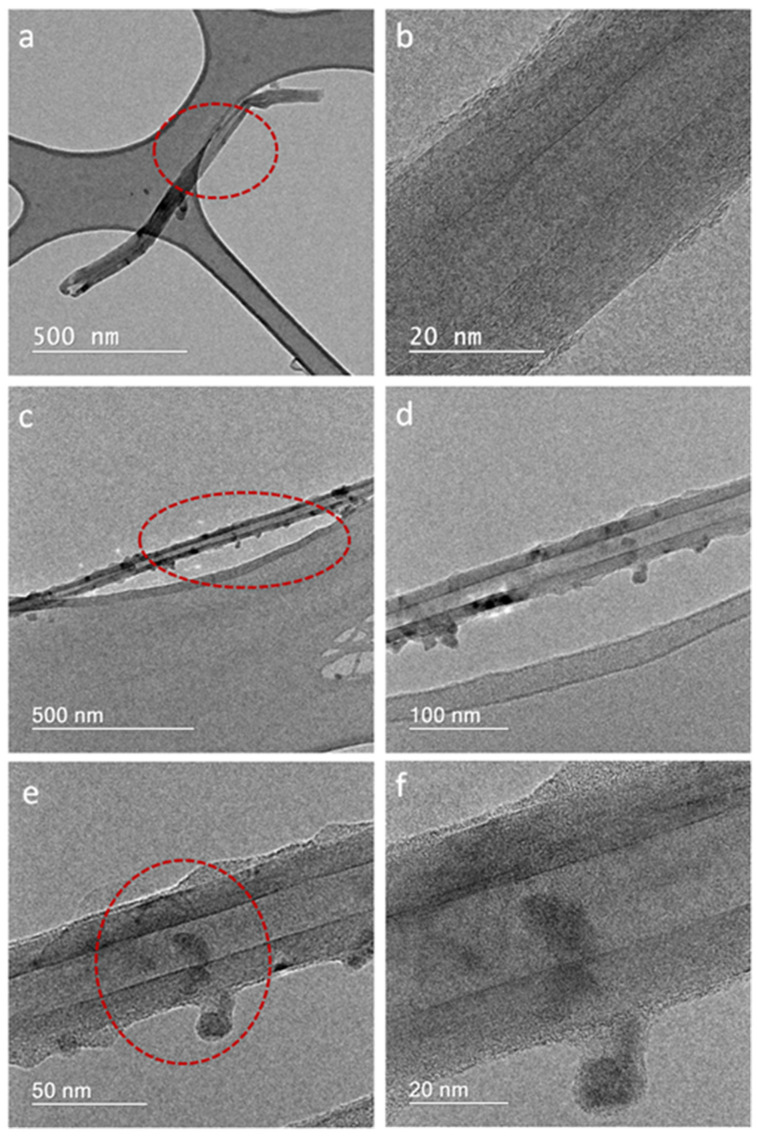
TEM images of BNNTs (**a**,**b**) and Cu-BNNTs (**c**–**f**).

**Figure 9 nanomaterials-11-02907-f009:**
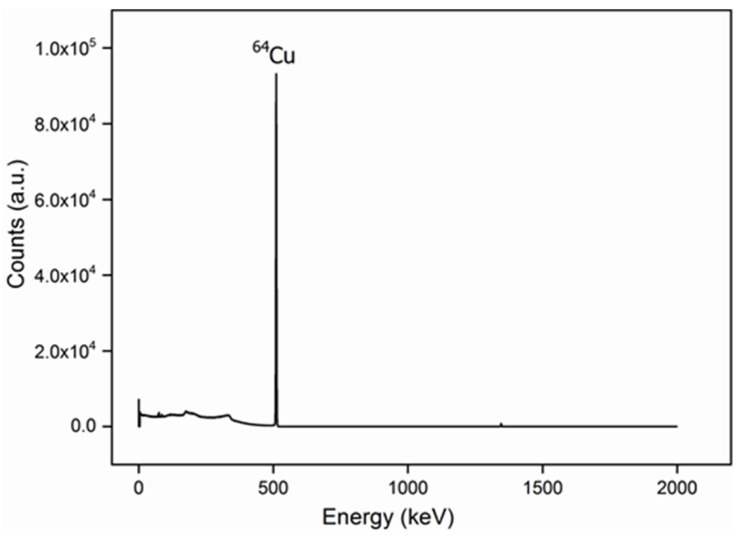
Gamma spectrum of ^64^Cu-BNNTs.

**Table 1 nanomaterials-11-02907-t001:** Surface composition (at.%) and B:N ratio, as determined by XPS for Cu-BNNT sample.

Elements	Bending Energy (eV)	Atomic Percentage (at.%)
O 1s	532.51	16.047
C 1s	284.49	26.010
N 1s	398.12	21.914
F 1s	689.51	4.179
Cu 2p	936.50	2.771
Cl 2p	198.53	3.004
B 1s	190.10	23.936
Si 2p	102.53	2.138
B:N	-	1.090

## Data Availability

Not applicable.

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
