# Peer review of "Potential Production of Theranostic Boron Nitride Nanotubes (64Cu-BNNTs) Radiolabeled by Neutron Capture"

_nanomaterials, 2021, doi:10.3390/nano11112907_

Round 1

Reviewer 1 Report

The authors reported the preparation of boron nitride nanotubes that was radiolabeled 64Cu by neutron capture. A series of characteristics have been carried out to demonstrate the successful production of the nanotubes. Parts of the work are interesting, however I am not sure that there is real justification for publication in nanomaterials, particularly due to the lack of application evidence in the biomedical field. Additionally, I think there are some relatively issues that should be addressed before publication elsewhere.

1: Since there have many commercial contrast agents and synthetic 64Cu nanomaterials reported for PET imaging, what are the main advantages of the nanotubes compared to those candidates?

2: As the authors have stated that the constructed nanotubes with appreciable properties can prove numerous multifunctional applications for cancer diagnosis and therapy, no experimental data can be found to support this statement. It is suggested to provide in vivo experiment to validate its theranostic potential.

3: There are many grammatical errors in the manuscript, for example, line 4 in the abstract, “observed” should be changed to “observe”; paragraph 4 line 2 in the introduction, “cam proved” should be “can prove”, etc. It is suggested to re-check the whole manuscript and correct all the mistakes.

Author Response

First, we thank the Reviewer 1 for bringing out the importance of our manuscript and recommending it for possible publication after revision.

REVIEWER 1

1: Since there have many commercial contrast agents and synthetic 64Cu nanomaterials reported for PET imaging, what are the main advantages of the nanotubes compared to those candidates?

We appreciate your comments. Contrast agents used in imaging exams are produced with substances (such as barium sulfate, iodinated contrast, or gadolinium, for example) capable of absorbing ionizing radiation and generating images. Despite its benefits, the use of contrast for exams contains risks, mainly of causing side effects such as allergic reactions, drop in blood pressure or intoxication of the kidneys and heart, for example, so they should only be used in specific cases, with adequate medical indication.

Although the study of 64Cu-BNNTs is still at an early stage, studies by Ciofani et al. (Ref. 10) indicate that this nanoplatform has cytotoxicity at concentrations below 50 µg/mL. Furthermore, BNNTs stabilize copper nanoparticles on their surface and in internal cavities, a fact observed in TEM images. Other expected advantages of this nanoplatform over contrast agents are: increased bioavailability, reduction of systemic adverse effects, increased patient comfort, adherence to treatment, improved osteogenic differentiation response promoted by 64Cu-BNNTs system and targeting to tumor cells. These advantages are described in the manuscript in the last paragraph of the introduction (page 2).

2: As the authors have stated that the constructed nanotubes with appreciable properties can prove numerous multifunctional applications for cancer diagnosis and therapy, no experimental data can be found to support this statement. It is suggested to provide in vivo experiment to validate its theranostic potential.

We thank the reviewer and agree with the comments. In vivo tests would be encompassed in the idealistic study, but in Brazil, the norms for the use of animals in scientific experiments are established by CONCEA (National Council for the Control of Animal Experimentation), an agency of the Ministry of Science, Technology, Innovation and Communications). Therefore, it is necessary for a project to be submitted and approved by this council. The project was sent, but due to the sanitary conditions imposed by the COVID-19 pandemic, the project is still under analysis. Below is the experimental protocol that will be adopted after the board's approval.

Experimental Protocol - in vivo biodistribution study

The in vivo studies will be conducted in female Balb/c mice (N = 40), with an average body mass of 20-25 g and aged 6-8 weeks. The animals will be kept in an area with light control and will have free access to water and feed. The mice will be housed in plastic cages containing wood shavings, respecting the limit of 5 (five) animals per box.

From a viable suspension of tumor cells (4T1) previously cultured, with an average density between 1.0 x 106 and 2.5 x 106 cells/mL, it will be injected into each animal for tumor development (N=20). A volume of 100 µL of this suspension will be implanted in the right flank of mice. The animals will be kept in an area with light control and will have free access to water and feed. About five days after cell implantation, the tumor will be visible and palpable. During the period of tumor growth, the animals will be monitored daily to check activity levels, mobility, and onset of cachexia, with the use of analgesia if necessary. On the sixth day, the animals will fast for 4 to 6 hours to start the biodistribution test. Therefore, a volume of radiolabeled nanoparticles suspension equivalent to an activity of 30 to 40 µCi will be injected into the tail vein of the animals. At predetermined times of 1, 2, 3, 4 and 5 hours (5 groups) after the injection of the nanoparticles, the mice (n =4 for each group) will be anesthetized intraperitoneally with a solution of Ketamine and Xylazine (80 mg/kg and 15 mg/kg, respectively) and sacrificed by cervical dislocation. Aliquots of blood and organs such as lung, liver, spleen, kidney, muscle, bone, heart, pancreas, intestines, bladder, and tumor tissue will be removed, washed, weighed and placed in plastic tubes for radioactivity counting. Blood will be collected with a needle and syringe, by cardiac puncture, soon after the animal's euthanasia. Like blood, urine will also be collected after euthanasia, with a needle and syringe. This collection will be carried out if, when opening the animal's abdomen to collect the organs, the presence of urine is found in the animal's bladder.

Two other groups of animals with N = 4 will be used to obtain PET images. Each group will receive the same dose of radiolabeled nanoparticles, but they will be imaged at different times that will be defined after the results of the biodistribution test. This procedure will allow us to analyze the potential diagnosis of the studied material and will allow us to develop an optimization plan for obtaining images. The immobilization of the animal will be done by placing it on a rough or gridded surface, with the tail held by the researcher (right hand) so that, with the other hand (left) the skin of the nape (just below the ears) is trapped between the index finger and thumb. Then the tail is placed between the 4th and 5th fingers. The mouse should be immediately turned over, with its paws up. The drug isoflurane 2-5% oxygen will be applied by inhalation as an anesthetic for immobilization of animals.

In vivo studies with healthy Balb/c mice (N = 20) will follow the same procedures mentioned above, excluding the tumor cell inoculation procedure

3: There are many grammatical errors in the manuscript, for example, line 4 in the abstract, “observed” should be changed to “observe”; paragraph 4 line 2 in the introduction, “cam proved” should be “can prove”, etc. It is suggested to re-check the whole manuscript and correct all the mistakes.

We appreciate your comments. We have corrected these grammatical errors in the manuscript.

Reviewer 2 Report

This work is interesting in BNNT community. The authors prepared and characterized BNNTs incorporated with 64Cu, which hold great potential for bioapplications. The most analysis was well performed with reasonable explanation. However, some characterization of materials was not fully provided. Thus, this reviewer suggests the publication in Nanomaterials after the major revision. My comments are below.

  1. Authors mentioned that –OH groups were introduced into BNNTs during the purification. They should provide IR spectrum and XPS data before and after the purification to confirm the formation and amount of –OH. In addition, the covalent functionalization of BNNTs may make BNNTs decomposed. Therefore, SEM data before and after the purification is also required.
  2. The explanation regarding the mechanism for the incorporation of Cu nanoparticles to BNNTs is required in the main text. Especially, I am wondering the initial nucleation origin. Does this result indicate the existence of numerous defects in BNNTs? or Is it due to hydroxyl group? The suggested mechanism should be supported by XPS and TEM results.

Author Response

REVIEWER 2

This work is interesting in BNNT community. The authors prepared and characterized BNNTs incorporated with 64Cu, which hold great potential for bioapplications. The most analysis was well performed with reasonable explanation. However, some characterization of materials was not fully provided. Thus, this reviewer suggests the publication in Nanomaterials after the major revision. My comments are below.

  1. Authors mentioned that –OH groups were introduced into BNNTs during the purification. They should provide IR spectrum and XPS data before and after the purification to confirm the formation and amount of –OH. In addition, the covalent functionalization of BNNTs may make BNNTs decomposed. Therefore, SEM data before and after the purification is also required.

We would like to thank the reviewer for his contribution, but we do not agree to add these results to the manuscript, because these data have already been published in previous works. Below is a more detailed explanation:

In preliminary works by our research group (Ref. 22), we identified through the XRD and TEM technique the presence of cubic iron at Bragg angle values ​​at around 44.8°. This occurs due to the formation of the reducing atmosphere at 1200 °C, where the Fe+2 ions present in the catalyst are reduced to metallic form. Therefore, for biological applications a standard purification procedure was adopted. This procedure is described in item 2.2 of the manuscript and in the Ref. 20. From this last reference, we identified by TGA analysis that the introduction of hydroxyl groups in defect sites in the tube structure is 4% and from XPS the values of the atomic percentage of hydroxyl groups introduced into the tubes. Because of these previous studies referenced in the manuscript, we believe that it is not important to carry out the comparative discussion between BNNTs (as grown) and purified BNNTs (BNNTs-OH) again.

  1. The explanation regarding the mechanism for the incorporation of Cu nanoparticles to BNNTs is required in the main text. Especially, I am wondering the initial nucleation origin. Does this result indicate the existence of numerous defects in BNNTs? or Is it due to hydroxyl group? The suggested mechanism should be supported by XPS and TEM results.

Through the XPS analysis, we observed that the relation in atomic percentage between the atoms of boron and nitrogen is of 1.09, in the other words, the composition of boron the structure is bigger than that of nitrogen leads to the formation of ternary BNxOy species, such as, BNNT-OH. Hydroxyl groups increase in modulus to negative charges on tubes surfaces. As copper nanoparticles have their surfaces positively charged, we believe that an electrostatic interaction occurs between the two nanostructures. Previous works by our research group (Ref. 12 and 19) using other nanoparticles reinforces this statement.

Note: The text in italics has been added to page 15 of the manuscript.

Round 2

Reviewer 1 Report

Based on the comments of the reviewers, the authors have perfected the work, and I recommend it for publication.

Reviewer 2 Report

I am satisfied with the response and recommend the publication in Nanomaterials.